# Towards using Tweet sentiment for infectious disease detection

**James Stassinos[1], Taylor Anderson[1]\*, Andreas Züfle[2]**

**1** Department of Geography and Geoinformation Science, George Mason University, Fairfax, Virginia, United States of America, **2** Department of Computer Science, Emory University, Atlanta, Georgia, United States of America

\* tander6@gmu.edu

**Data availability statement:** Data underlying the results presented in the study area available from CrisisNLP at https://crisisnlp.qcri.org/tbcov and USA Facts

## Abstract

Social media data has shown potential for identifying infectious disease outbreaks faster than official records of disease incidence. We examine spatial, temporal, and spatiotemporal relationships between COVID-19-related microblog sentiment and COVID-19 cases over space and time to investigate whether microblog-derived sentiment can be used for local infectious disease outbreak early warning. Therefore, we measure the sentiment of 56,755,894 COVID-19 related microblogs (tweets) from the microblogging platform X. We group these tweets by county and by calendar week to investigate spatial and temporal correlation between sentiment and observed cases (in the corresponding county and week). Our temporal analysis shows a significant negative correlation between sentiment and cases between June and September 2020. During this time, tweet sentiment could have served as an early warning for new COVID-19 outbreaks. Our spatial analysis shows that the East of the United States exhibits a significant negative correlation between Sentiment and Cases while the West exhibits a significant positive correlation. In these regions, Tweet Sentiment could have been used as an early warning signal for new outbreaks. Our spatiotemporal analysis discovers even stronger correlations in certain regions during certain time periods. If we could understand when, where, and why this correlation is strong, then we may be able to leverage social media as a successful early warning system.

## Introduction

Accurate and timely data capturing the ecology and evolution of infectious diseases such as COVID-19 has been crucial for informing policy interventions [1–3]. To measure the cases and deaths due to COVID-19 across the US, the Center for Disease Control (CDC) gathered data from jurisdictional and state partners that independently report the new daily number of confirmed COVID-19 cases and deaths for each county of residence based on testing data captured by pharmacies, hospitals, and other testing facilities [4]. However, due to a decentralized and fragmented public health infrastructure, this data is subject to inherent biases due to budget and resource availability or group characteristics, under-ascertainment of mild cases, the changing definition of a "confirmed case," reporting lags, and disentanglement of

https://static.usafacts.org/public/data/covid-19/covid_confirmed_usafacts.csv. CrisisNLP makes available the TBOV dataset and sentiment analysis. USA Facts makes available county level data capturing COVID-19 prevalence. Data processing scripts are available through the following GitHub repository https://github.com/jstassinos/Processing-the-TBCOV-Dataset.

**Funding:** This study was funded by the National Science Foundation (NSF) Awards #2109647, #2302968, and #2302970.

**Competing interests:** The authors have declared that no competing interests exist.

the cause of death [5–7]. One such example is testing bias, where some locations may have better testing infrastructure, well-funded access to testing, and less stigma around getting tested [5]. In another example, a lack of standardization resulted in an inconsistent set of reporting metrics where some states defined a "confirmed case" as a total count of positive tests and others as the total number of unique individuals that tested positive [8]. Furthermore, delays and backlog in the reporting pipeline resulted in a 3-21 day lag between the time a patient tested positive and reported as having tested positive [9].

Publicly available data from news outlets, chat rooms, web searches, or social media have been proven to be a valuable tool for disaster response and management, providing insights into both real and perceived threats [10,11]. For example, Kryvasheyeu et al. [12] find strong relationships between the proximity to tornadoes, hurricanes, earthquakes and other natural disasters and Twitter activity. Such data has also been used as a supplement to official data sources to identify disease outbreaks faster than what is reported by the CDC and can even detect outbreaks not detected by official sources [13–16]. Collectively, these sources, referred to as digital health data, provide a lens into public health that is fundamentally different from that yielded by official sources [17]. Among such sources, microblogs such as those posted by users on X (the platform formerly known as Twitter) are used to detect the prevalence of diseases such as influenza and dengue fever. In some cases, the volume of such microblogs (called Tweets) is used as an indicator of local disease outbreaks. More specifically, the number of tweets that contain keywords relating to a disease or have been classified as a "self report" have been found to correlate with CDC reported cases and other official sources and thus can be used as a tool for early detection and monitoring [18–23].

In other cases, additional analyses of the Tweets, using natural language processing approaches such as sentiment analysis, have been used to differentiate between Tweets that are positively, i.e. "my flu shot worked, no flu for me!" and negatively associated with disease, i.e. "my whole family has the flu" and predict to what extent influenza is present in the population over time [24–27]. Where Tweet sentiment and official influenza cases tend to be inversely associated, the relationship is less clear for COVID-19. For example, Valdez et al [28] were surprised to observe a positive correlation between US wide COVID-19 related Tweet sentiment and cases and deaths, meaning that as cases and deaths increase in the US, sentiment towards COVID-19 trends positive. This contradicts what would intuitively be expected. In another example, Feng and Kirkley [29] find a weak negative or absent correlation between state-level Tweet sentiment and COVID-19 and cases and deaths.

Therefore, the objective of this study is to examine the relationship between COVID-19 related Tweet sentiment and official case data over space and time and to assess the extent to which Tweet-derived sentiment can be used for local COVID-19 surveillance in the United States. We leverage the TBCOV dataset [30], containing a total of two billion geolocated COVID-19 related Tweets between Jan 2020 to Dec. 2021. After pre-processing, we use approximately 57 million geo-located tweets in the United States. Details of the pre-processing are described in the Data section. We examine the county-level correlation between Tweet sentiment and official data from February 1, 2020 to March 31, 2021 to determine whether, when, and where Tweet sentiment from a county can be used as a predictor of the number of cases in the same county. To extract sentiment for Tweets we use existing sentiment analysis algorithms which leverage natural language processing, text analysis, and computational linguistics to systematically identify, extract, quantify, and study people's moods, opinions, attitudes, and emotions in written text [31]. For each Tweet, we measure the Sentiment Polarity, which is a score associated with a direct opinion about an object on a scale from –1 (negative) to +1 (positive) [31]. We leverage multiple sentiment analysis tools including as Text

Blob [32], Vader Sentiment [33], AFINN [34] and TBCOV [30] as our measures for Sentiment Polarity.

We intuitively assume that there should be an inverse association between COVID-19 Tweet sentiment and COVID-19 cases at the local level. Thus, places experiencing a high number of COVID-19 cases are expected to have a low COVID-19 sentiment during that time. We can observe this inverse association in Fig 1 which shows both the COVID-19

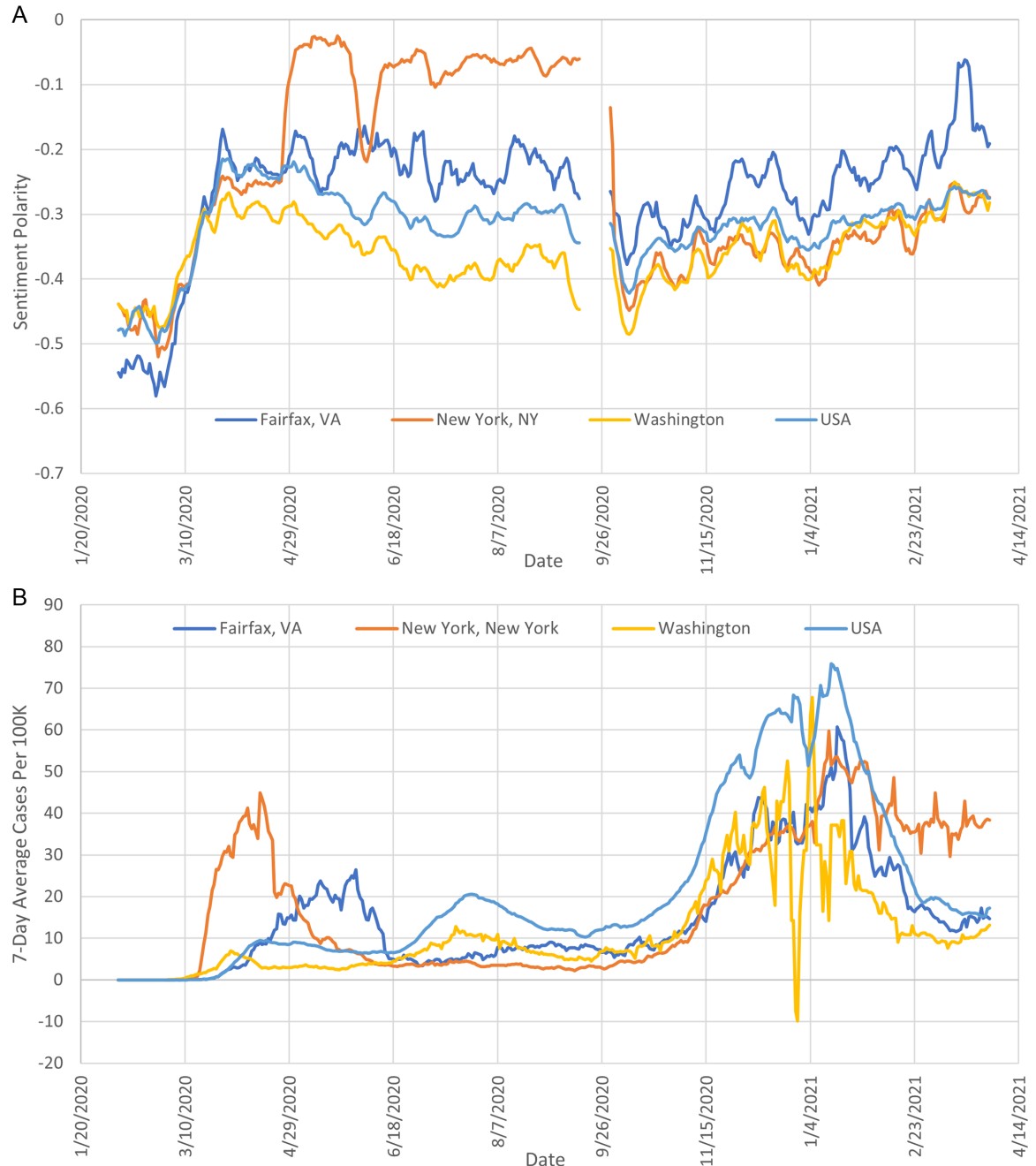

**Fig 1. Seven-day rolling average of (a) COVID-19 tweet sentiment and (b) COVID-19 incident cases.**

related sentiment in Tweets and the number of COVID-19 cases for the entire U.S. and for some locations in the U.S. For example, in the case of New York City, near the end of April 2020, when number of local COVID-19 cases dramatically decreased, we observe a large increase in mean sentiment. Yet, in general, it is difficult to discern any clear relationship, thus, warranting further investigation. In the case that there is an inverse association between COVID-19 Tweet sentiment and COVID-19 cases at a local level, we anticipate that COVID-19 related Tweet sentiment could be used to supplement official disease surveillance data streams and provide important insights for local outbreak detection of diseases [35]. To our knowledge, there is no existing study that examines the relationship between Tweet sentiment and infectious disease cases at a spatially local level.

## Data

### Tweet preprocessing

The primary source of geo-located Tweets used in this work is the TBCOV dataset [30]. This dataset contains over two billion Tweets collected world-wide with keywords related to COVID-19. All Tweets are enriched with geo-location information, using either the location directly provided by the user or using location information from publicly available user profiles. The dataset covers a 424 day period from February 1,2020 to March 31, 2021. Since the X platform (formally Twitter) does not allow the re-distribution of data from Tweets, the TBCOV dataset is provided as a "dehydrated" dataset which includes only the Tweet identifiers. To obtain the full dataset, we rehydrated the Tweets using the Twitter API (now with reduced capabilities).

TBCOV is a global dataset. Therefore, we first removed all of the Tweets from outside of the US, reducing the number of tweets that we consider to 384,073,303. To avoid redundant information, we next removed all re-Tweets from the dataset. Re-Tweets echo the tweet of another user and would contribute noise to our analysis. This step reduced the number of Tweets to 120,678,732. Furthermore, some Tweets were not successfully rehydrated because the original Tweets may have already been removed—either through deletion by the user or because an account has been removed or banned from the platform. The Tweets in this research were hydrated through April 3, 2022 to April 10, 2022. Depending on when a person hydrates the dataset the number of Tweets will change. Out of the 120,678,732 unhydrated Tweets, we were successfully able to rehydrate 77% of Tweets for a total of 93,362,576. Globally, very few Tweets (0.1%) in the TBCOV dataset have explicit coordinates (latitude, longitude). Thus, for the majority of the Tweets, the Tweet location is inferred either based on the user location or the content of the Tweet text. The authors of the TBCOV dataset report 76.1% and higher F1-scores for county-level localization using the location of the user [30]. However, for the case of locations inferred from tweet text, the F1-score was only 0.100%. For this reason, we discarded all tweets whose location is estimated using tweet text. After removing such Tweets, our dataset was reduced to a total of 56,755,894 Tweets. The TBCOV dataset was able to infer the user type of the tweets based on associated account information. We chose to filter down to just the user types labelled as a person, to improve our ability to detect a persons emotion among the other user types on the platform. After filtering for people the dataset was reduced to 30,328,442 tweets. Using our set of Tweets that are reliably geolocated, we then group all Tweets by day and by county. By doing so, we observe that many counties have too few observed Tweets to be a representative sample Therefore, we used a 7-day rolling sum of Tweets for each county and discarded (county, day) pairs with fewer 30

Tweets total across the seven days. The original TBCOV dataset has a gap in the tweet collection from September 15, 2020, to September 23, 2020. This period plus the 7 days following were omitted from the final aggregation.

## Sentiment measures

Sentiment is generally measured on a scale from –1 to + 1 where the signum describes the polarity (+, –) and the absolute describes the magnitude of the sentiment. To compute the sentiment of Tweets for a (county, day) pair, we apply four common methods of calculating sentiment from text. These are Text Blob [32], Vader Sentiment [33], Afinn [34], and an included calculation denoted as TBCOV Sentiment [36].

TextBlob [32] is a commonly used natural language processing module for Python. To estimate Sentiment Polarity, TextBlob utilizes a rule-based system for sentiment analysis, which involves a predefined list of words with associated sentiment polarities. For this list, we use the commonly used list sourced from the Python Natural Language Toolkit (NLTK) database. TextBlob returns a score between 1 and –1 indicating both polarity (positive or negative) and strength of the sentiment [32].

Valence Aware Dictionary for Sentiment Reasoning (VADER) is the second Python module that will be used to calculate sentiment on the Tweets. It is a lexicon-based sentiment analyser. This tool is also built on top of NLTK. It calculates the polarity of sentiment by matching sentiment intensity scores to words and then aggregating these scores into an overall score. This process is repeated for each Tweet in the database. VADER returns a number between 1 and –1 for polarity [33].

Afinn is the third sentiment calculation Python module named after its developer Finn Årup Nielsen. The developer created a dictionary of 2477 words for calculating polarity scores. Afinn is a lexicon-based sentiment analyzer. It returns a number between 5 and –5 for polarity [34].

TBCOV Sentiment uses a multilingual transformer-based deep learning model. This returns a polarity number between 1 and –1, and a confidence score. It uses a transformer-based deep learning model called XML-T, trained on millions of general-purpose tweets in 8 different languages [36]. Figure **??** shows the sentiment polarity from TBCOV from February 1, 2020 to March 31, 2021. We present the results of our analysis using the TBCOV sentiment in this paper with the results using the other sentiment measures in the Supplemental Materials.

## COVID-19 case data

The number of new confirmed COVID-19 cases per county over time, also known as incident cases, is collected by the Center for Disease Control (CDC) and is published by USA Facts [37]. To smooth the case data, a 7-day rolling average is applied. To account for population differences the daily case counts are normalized by population. Thus, when we refer to COVID-19 cases from this point on, we mean a 7-day rolling average of incident cases normalized by population to a per 100,000. Fig **??** shows the daily COVID cases from February 1, 2020 to March 31, 2021.

## Sentiment-disease case correlation analysis

To examine the relationship between COVID-19 Tweet sentiment and official case data over space and time and to assess the extent to which Tweet-derived sentiment can be used for

local COVID-19 surveillance in the United States, we first group the observed COVID-19-related tweets by calendar week and county and compute the average sentiment of tweets in this group. This grouping and aggregation provides us with one set of Tweets for each county $r$ (among all counties $\mathcal{R}$) and for each week $w$ (among all weeks $\mathcal{W}$). Deriving the sentiment of each such set provides us with a function $S(w,r)$ that returns the sentiment of all tweets observed in region $r$ during week $w$. In the following, we define measures to quantify the correlation, globally, spatially, and temporally, between these sets $S(w,r)$ and the number of COVID-19 cases $C(w,r)$ observed in the same region during the same week.

**Global analysis.** To measure the correlation between COVID-19 Tweet sentiment and COVID-19 cases observed over all weeks and all regions, we simply measure Pearson's correlation across space and time, as follows:

$$\mathrm{corr}(\mathcal{W},\mathcal{R}) = \mathrm{corr}_{w \in \mathcal{W}, r \in \mathcal{R}}(S(w,r), C(w,r)) = $$
$$\frac{\sum_{w \in \mathcal{W}, r \in \mathcal{R}}(S(w,r) - \overline{S}) \cdot (C(w,r) - \overline{C})}{\sqrt{\sum_{w \in \mathcal{W}, r \in \mathcal{R}}(S(w,r) - \overline{S})^2 \cdot \sum_{w \in \mathcal{W}, r \in \mathcal{R}}(C(w,r) - \overline{C})^2}}, \quad (1)$$

where $\overline{S}$ denotes the average sentiment (across all weeks and counties) and $\overline{C}$ denotes the average number of cases (across all weeks and counties). Equation 1 provides us with the sample correlation coefficient (a single scalar) that measures the correlation between COVID-19 Tweet sentiment and COVID-19 cases. Intuitively, we would expect a negative correlation as a high number of cases should (in average across all weeks and counties) yield a low (negative) sentiment towards COVID-19. However, our experimental evaluation shows that this intuition is not confirmed by the data. That is because an aggregation across all weeks and all counties overgeneralizes any interesting spatial and temporal patterns. To find such patterns, we next propose to measure the correlation between COVID-19 Tweet sentiment and COVID-19 disease cases both temporally local (for a specific "frozen" week in time) and spatially local (for a specific "frozen" county).

**Temporal analysis.** To understand whether and how the correlation between COVID-19 sentiment and COVID-19 cases changes over time, we define a by-week correlation over all spatial regions $\mathcal{R}$, as follows:

$$\mathrm{corr}(w) = \mathrm{corr}_{r \in \mathcal{R}}(S(w,r), C(w,r)) = $$
$$\frac{\sum_{r \in \mathcal{R}}(S(w,r) - \overline{S(w)}) \cdot (C(w,r) - \overline{C(w)})}{\sqrt{\sum_{r \in \mathcal{R}}(S(w,r) - \overline{S(w)})^2 \cdot \sum_{r \in \mathcal{R}}(C(w,r) - \overline{C(w)})^2}}, \quad (2)$$

where $\overline{S(w)} = \frac{\sum_{r \in \mathcal{R}} S(w,r)}{|\mathcal{R}|}$ is the average sentiment across all counties during week $w$ and $\overline{C(w)} = \frac{\sum_{r \in \mathcal{R}} C(w,r)}{|\mathcal{R}|}$ is the average number of cases across all counties during week $w$.

Equation 2 provides us with the sample correlation coefficient between sentiment and cases across all states during week $w$. Intuitively, we would expect that this correlation should be stationary over time, such that at any time a region having a higher number of cases should have a lower sentiment towards COVID-19. We will evaluate this hypothesis by computing $\mathrm{corr}(w)$ for each week and plotting the resulting time series (and corresponding p-values and z-scores of the significance of the correlations) in our experiments results section.

**Spatial analysis.** To understand whether the correlation between COVID-19 sentiment and COVID-19 cases is stationary over space, we compute the correlation across all weeks $\mathcal{W}$ of our dataset for each spatial region $r$. This correlation is defined as:

$$\text{corr}(r) = \text{corr}_{w \in \mathcal{W}}(S(w,r), C(w,r)) = \tag{3}$$

$$\frac{\sum_{w \in \mathcal{W}}(S(w,r) - \overline{S(r)}) \cdot (C(w,r) - \overline{C(r)})}{\sqrt{\sum_{w \in \mathcal{W}}(S(w,r) - \overline{S(r)})^2 \cdot \sum_{w \in \mathcal{W}}(C(w,r) - \overline{C(r)})^2}}, \tag{4}$$

where $\overline{S(r)} = \frac{\sum_{w \in \mathcal{W}} S(w,r)}{|\mathcal{W}|}$ is the average sentiment in region $r$ over all weeks and $\overline{C(r)} = \frac{\sum_{w \in \mathcal{W}} C(w,r)}{|\mathcal{W}|}$ is the average number of cases across in region $r$ over all weeks.

Equation 3 provides us with the sample correlation coefficient between sentiment and cases observed in region $r$ during the entire COVID-19 pandemic. Intuitively, we would expect that this correlation should be stationary over space, such that, for all regions, we'd expect that times having a higher number of cases should have a lower sentiment towards COVID-19. We will evaluate this hypothesis by computing $\text{corr}(r)$ for each region $r$ and mapping the results across the United States in our experiments results section.

**Spatiotemporal analysis.** In order to understand whether the correlation between COVID-19 sentiment and COVID-19 cases is stationary over space and time, we divide the entire temporal extent of the data (March 02, 2020 - March 31, 2021) into three temporal intervals: wave 1 (March 02-June 07, 2020), wave 2 (June 9–August 24, 2020), and wave 3 (August 26–December 30, 2020) and perform spatial analysis for each period.

## Results

This section presents the results of our correlation analysis between COVID-19-related Tweet sentiment and COVID-19 cases globally (Equation 1), across time (Equation 2), and across space (Equation 3). All results shown in the following use the TBCOV Sentiment Polarity measure. The results for TBLOB, VADER, and AFINN Sentiment Polarity are similar and are presented in Supplementary material.

### Global correlation

Table 1 shows the global correlation between COVID-19 Tweet sentiment and COVID-19 cases using the four different sentiment measures TBCOV [30], TBLOB [32], VADER [33], and AFINN [34].

Surprisingly, we observe that the negative correlation that we would intuitively expect only holds using three out of the four sentiment measures. For VADER, we observe a positive

**Table 1. Statistical values for overall correlation.**

| Sentiment | Correlation coefficient | T-Stat | P-Val |
|---|---|---|---|
| TBCOV | -0.05745 | -44.12266 | 0.00000 |
| TBLOB | -0.06347 | -48.76026 | 0.00000 |
| VADER | 0.01663 | 12.75548 | 0.00000 |
| AFINN | -0.02421 | -18.56581 | 0.00000 |

correlation. For all the others, we do observe a negative correlation. The magnitude of the correlation is weak ($\leq 0.016$). Despite the weak correlations, all these correlations are highly significantly different from zero (using a Student's t test) due to the very large number of (week, county) pairs.

> To summarize our global correlation analysis, we observe some significant correlation, but the magnitude is weak and even the direction of the correlation differs between sentiment measures. This rather inconclusive result confirms observations made in prior work [28,29].

A

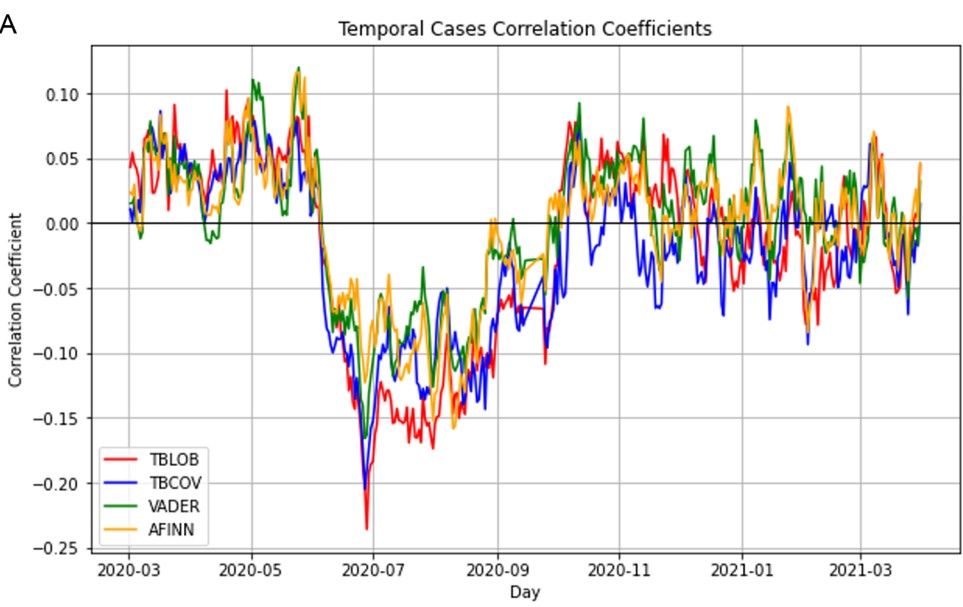

B

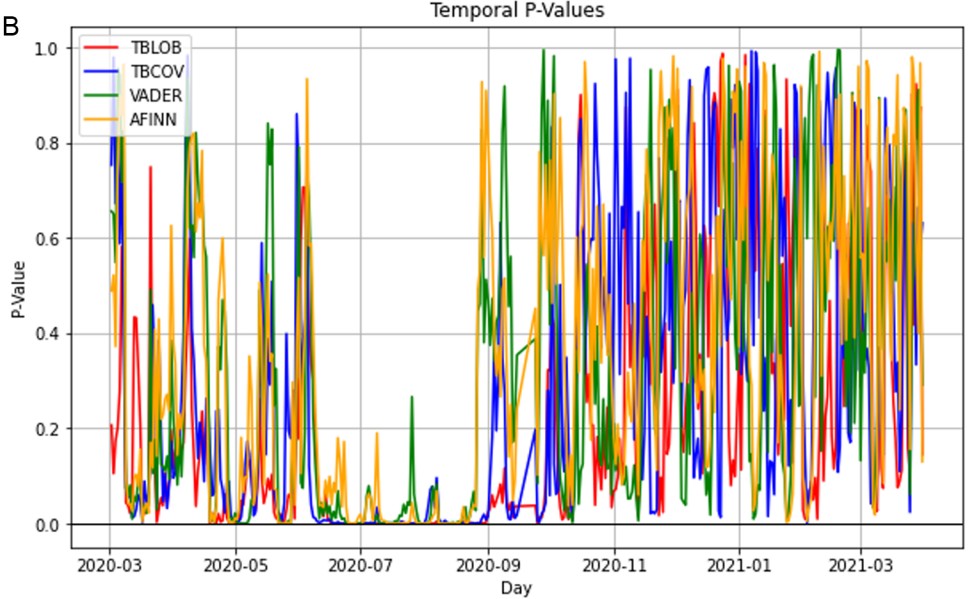

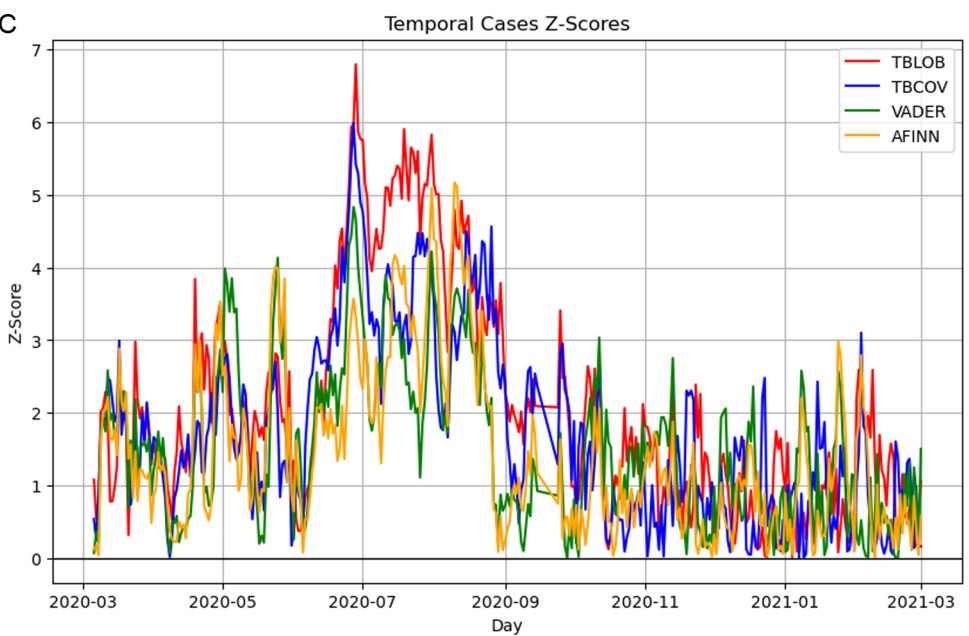

**Fig 2. (a) Daily correlation coefficient between sentiment measures and daily new cases per 100,000 in each county and (b) the corresponding p-values and (c) z-scores.**

To gain a better understanding of the link between tweet sentiment and COVID-19 cases, we next evaluate the hypothesis that correlations are space and time-homogeneous. The following will show that this hypothesis can be rejected at a high level of confidence as correlations are significant positive and strong in some regions during some time periods, but negative and strong in other regions during other periods.

## Correlation over time

Based on Equation 2, Fig 2a presents the correlation coefficient between a 7-day moving average of TBCOV sentiment and a 7-day moving average of COVID-19 cases for all counties from January 20, 2020 to April 14, 2021. First, we observe that the hypothesis of temporal homogeneity is not supported by the data: We observe that different time intervals exhibit different magnitudes and directions of correlation between Tweet COVID-19 sentiment and COVID-19 cases. Fig 2b also provides the p-values of each weekly correlation having p-values close to zero on days where the correlation (over the last seven days) was significantly different from zero. We also provide the corresponding z-scores in Fig 2c indicating the daily (over the last seven days) absolute standard deviations that the observed correlation coefficients deviate from zero.

Looking at Fig 2a more closely, we can see that four time intervals exhibit significantly different correlations. The first time interval is a period of having a weakly significant (p-values <0.05 on most days) positive correlation between tweet sentiment and COVID-19 from March 2, 2020 to June 10, 2020 where as COVID-19 cases increase, tweet sentiment increases. This unexpected positive correlation may be explained by having many counties that had zero cases at this time, but which did already exhibit a low sentiment towards COVID-19 despite the zero cases. The strongest and most significant result can be observed in the time interval from June 12, 2020 to August 15, 2020. During this period the correlation between sentiment

and COVID-19 cases abruptly shifts to a period of highly significant (p-values approaching zero) relatively strong (0⊠5 magnitude) negative correlation where as COVID-19 cases increase, tweet sentiment decreases. This short period aligns with our intuition that places with a high number of cases should have a negative sentiment towards COVID-19. We note that the p-values and z-scores shown in Fig 2 are computed independently for each day and the probability of having such low p-values over sixty days in a row are infinitesimal indicating a highly significant result. Third, we observe another shift to a period of positive correlation from November 2, 2020 to December 6, 2020 similar to the first period. Finally, from December 6 onward, the correlation coefficients fluctuate around a sentiment of zero with p-values seemingly uniform in the interval [0,1] indicating no more significant correlation after this time.

> As the main takeaway from this analysis, we observed a time interval in which the negative correlation between sentiment and cases was significantly and strongly correlated - indicating that during this time, tweet sentiment could have been successfully used as an early warning indicator.

## Correlation over space

Using Equation 3 to obtain a correlation measure for each county, Fig 3 presents the correlation coefficients map, aggregated over all 424 days. For this experiment, we omitted any county with fewer than 200 days of data and any day with less than 30 Tweets across a 7 day period. After omitting counties with insufficient data, we retained 1,442 counties out of all the 3,025 counties that have at least one tweet in the dataset. While correlations aggregated to the entire United States were rather weak as seen in our Global Correlation study, we first observe that some counties indeed exhibit strong correlations, ranging from −0.76 to 0.51. We also observe that these correlations exhibit spatial clustering, with many counties having either a strong negative correlation (red colour in Fig 3a) or medium negative correlation (orange colour) in the North Eastern US. Counties having positive correlations (light and dark blue in Fig 3a) are less frequent, which is expected as the average correlation for TBCOV is −0.055 as shown in Table 1, but appear more frequently in the West and Midwest. As these spatial trends may not be immediately evident from Fig 3a, we applied Anselin's test for local spatial autocorrelation to the correlations values obtaining the map of local clusters as shown in Fig 3b. As expected, we observe a large cluster of low correlation values to the Northeast (although having many outliers of relatively high values within this low cluster). We see a large cluster of high values (indicating positive correlation) in the West and Midwest with parts of California excluded from this cluster as not significant.

> The main result of the spatial analysis is that there are regions within the United States for which there exists a strong correlation (either negative or positive) between tweet sentiment and COVID-19 cases. This shows that in these regions, tweet sentiment could have been successfully used as an early warning indicator.

## Spatiotemporal analysis

The previous sections have shown that, despite the weak global correlation between tweet sentiment towards COVID-19 and actual COVID-19 cases across all weeks and regions, there are:

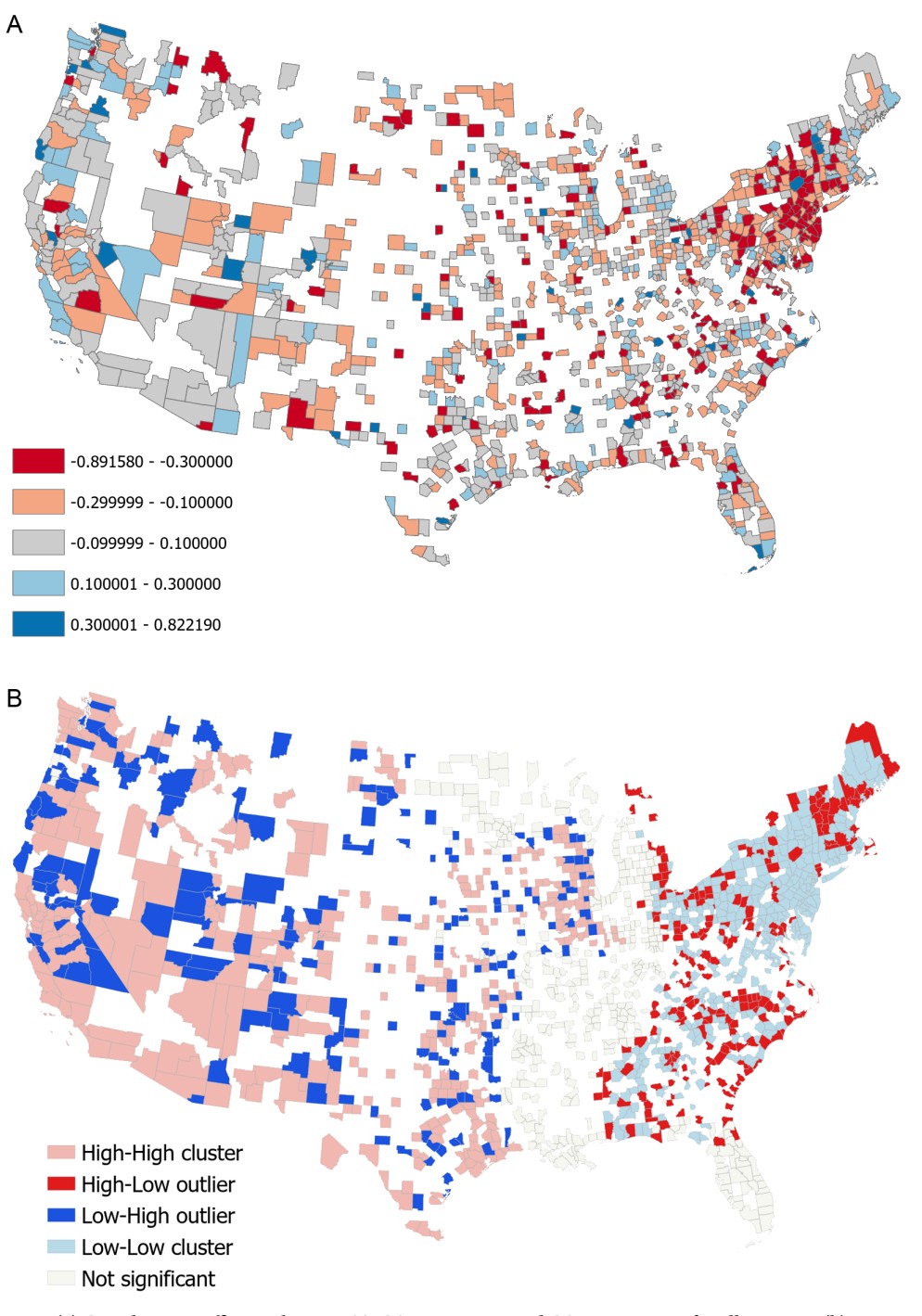

**Fig 3. (a) Correlation coefficients between TBCOV sentiment and COVID-19 cases for all counties. (b) Anselin local Moran's I cluster and outlier analysis of TBCOV correlation coefficients. An interactive version of this map can be found at https://mygmu.maps.arcgis.com/apps/mapviewer/index.html?webmap= e73b409615824f9f9e019623b42ae664.**

- Some periods of time when where the correlation between tweet sentiment and cases is relatively strong and highly significant and

- There are regions within the United States in which the correlation between tweet sentiment towards COVID-19 and actual COVID-19 cases is relatively strong (either positive or negative).

These observations raise the following question: Are there any combinations of time and region when and where these correlations are particularly strong? And could we use this knowledge for the prevention of future epidemics? This section answers these question by investigating the spatiotemporal correlation between sentiment and COVID-19 cases during distinct periods of the pandemic. Therefore, we select three specific time intervals corresponding to the three first waves of COVID-19 cases as observed in Fig 1: (1) The first wave from March 02, 2020 to June 07, 2020; (2) the second wave from June 09, 2020 to August 24, 2020; and (3) the third wave from August 26, 2020 to December 30, 2020. For each of these waves, we repeat the spatial analysis presented in the previous section. We report our results in Fig 4, where each map corresponds to a distinct period. For comparison, Figure 4a) shows the results of the spatial analysis over the entire time period which is identical to Fig 3a. The first wave from March 02, 2020 to June 07, 2020 shows an overall more positive correlation than the other periods. The spatial distribution reflects a predominantly positive relationship across regions. In contrast, the second period from June 09, 2020 to August 24, 2020 exhibits a noticeable shift towards a negative correlation between sentiment and cases. In the third period from June 09, 2020 to August 24, 2020, the overall correlation reverts to a positive trend, resembling the first period. Notably, the areas around New York and Philadelphia exhibit a distinct negative correlation. We can visually observe that the correlations in specific time periods (Fig 4b–4d) are much stronger than the correlations over the entire study period (Fig 4a).

> The main results of the spatiotemporal analysis and of this work is that there are certain times when certain regions exhibit a very strong correlation (either negative or positive) between tweet sentiment and COVID-19 cases. If we could understand and predict when and where these strong correlations occur, we could leverage this understanding to use tweet sentiment as an early indicator for cases.

### Explanation of correlations

Towards understanding when and where these correlations occur, in this section, we investigate the links between the observed correlations as seen in Fig 4 with population variables related to population density, socioeconomic differences, education, and medical resources. Table 2 shows the result of a regression analysis using the following population variables that we obtained from the County Health Rankings and Roadmaps Dataset [38]:

- Percentage of Population without a High School Degree,
- The number of primary care physicians per population,
- The percentage of population unemployed,
- The percentage of population of age 65 or older,
- The percentage of population living in a census-defined rural area.
- The population density, defined as population per square kilometre.

We perform this regression analysis separately for each of the four time-periods used in Fig 4:

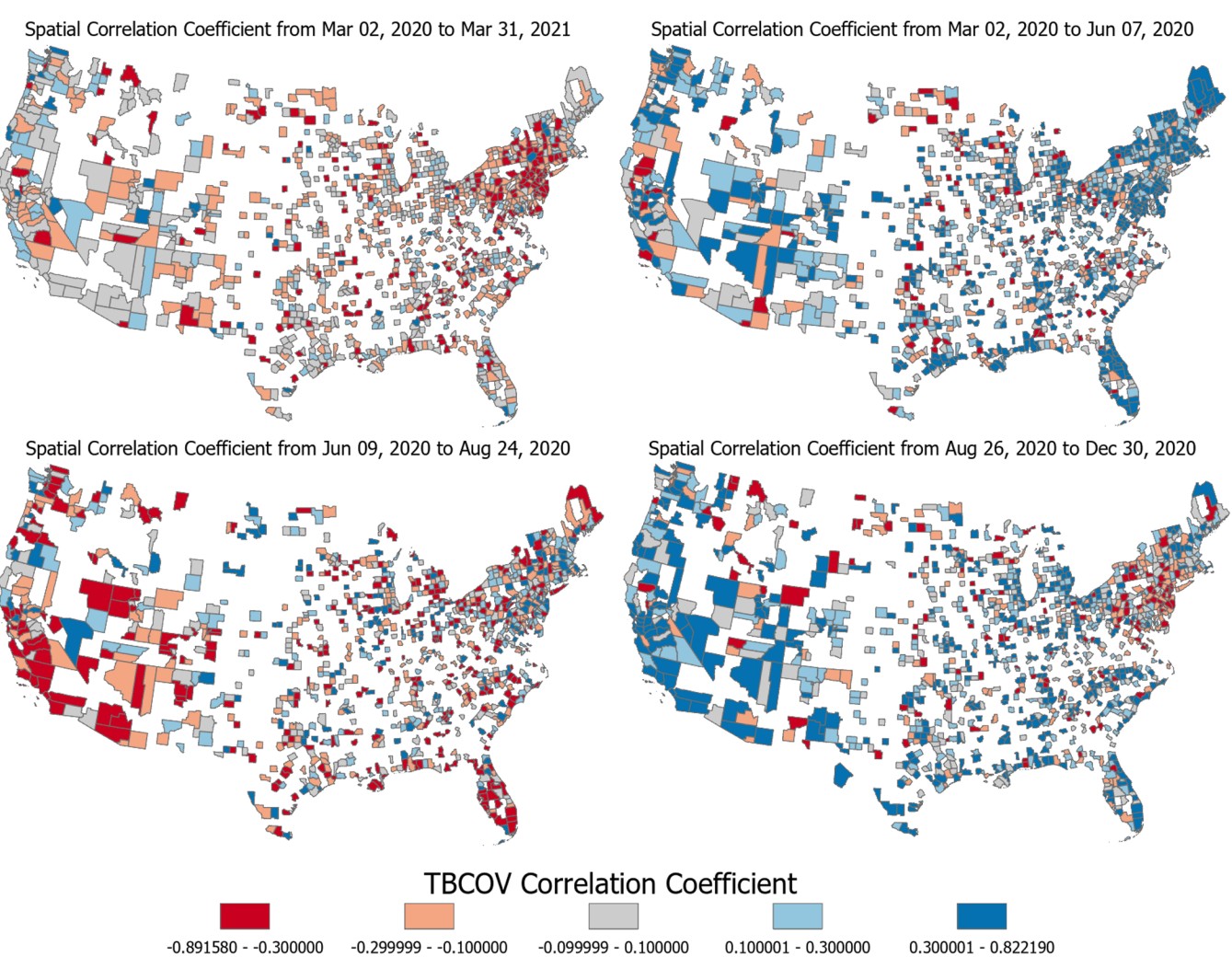

**Fig 4. Fig 4. TBCOV correlation coefficients visualized spatially for 3 different time periods. An interactive version of this map can be found at https://mygmu.maps.arcgis.com/apps/mapviewer/index.html?webmap=1214ac65df774fcda693fd794141574f.**

- The full duration including the first three COVID-19 waves from March 02, 2020 to March 31, 2021.
- The duration of the first wave from March 02 to June 07, 2020.
- The duration of the second wave from June 09 to August 24, 2020, and
- The duration of the third wave from August 26 to December, 2020.

First, we observe that, over the entire duration, these population variables do not explain the correlations between COVID-19 cases and Twitter sentiment well, shown by the low adjusted $R^2$ coefficient of determination of 0.002 for the entire duration. For the full duration, none of the above population variables are significantly correlated to the case-sentiment correlations at a level of significance of less than 3%. This result is consistent with the temporal analysis in Fig 2b, due to the signum of case-sentiment correlations changing over time from positive (Wave 1), to negative (Wave 2), back to positive (Wave 3), thus having no significant correlation during the entire period.

**Table 2. Regression results for different time periods.**

| Variable | Full Period | Wave 1 | Wave 2 | Wave 3 |
|---|---|---|---|---|
| | Mar-Dec | Mar 2-Jun 7 | Jun 9-Aug 24 | Aug 26-Dec 30 |
| Constant | -0.152 | 0.256 | -0.066 | 0.155 |
| std. | 0.035 | 0.050 | 0.063 | 0.054 |
| p-value | 0.000*** | 0.000*** | 0.293 | 0.004*** |
| Education (No HS) | -0.057 | -0.177 | -0.149 | 0.173 |
| std. | 0.153 | 0.216 | 0.270 | 0.237 |
| p-value | 0.711 | 0.413 | 0.581 | 0.467 |
| Primary Care per 100k | 0.0002 | 0.0006 | 0.00008 | 0.0007 |
| std. | 0.0002 | 0.0002 | 0.0003 | 0.0003 |
| p-value | 0.271 | 0.015** | 0.805 | 0.012** |
| Unemployment (%) | -0.053 | -0.262 | 0.423 | -0.893 |
| std. | 0.547 | 0.771 | 0.970 | 0.841 |
| p-value | 0.923 | 0.735 | 0.663 | 0.289 |
| 65 or Older (%) | 0.114 | 0.088 | -0.196 | -0.449 |
| std. | 0.151 | 0.214 | 0.261 | 0.227 |
| p-value | 0.450 | 0.681 | 0.453 | 0.048** |
| Rural (%) | -0.035 | -0.200 | 0.193 | -0.128 |
| std. | 0.027 | 0.038 | 0.048 | 0.042 |
| p-value | 0.194 | 0.000*** | 0.000*** | 0.002*** |
| Poverty (EP_POV150) | 0.002 | -0.002 | -0.003 | 0.003 |
| std. | 0.001 | 0.002 | 0.002 | 0.002 |
| p-value | 0.035** | 0.133 | 0.161 | 0.041** |
| Population Density | -8.22e-06 | 4.23e-06 | -1.35e-05 | -2.55e-05 |
| std. | 8.92e-06 | 1.27e-05 | 1.47e-05 | 1.38e-05 |
| p-value | 0.357 | 0.740 | 0.359 | 0.065* |
| Observations | 1,249 | 1,291 | 1,071 | 1,226 |
| R-squared | 0.008 | 0.065 | 0.023 | 0.037 |
| Adjusted R-squared | 0.002 | 0.060 | 0.017 | 0.031 |
| F-statistic | 1.401 | 12.76 | 3.654 | 6.604 |
| p-value (F-stat) | 0.201 | 0.000*** | 0.000*** | 0.000*** |

*Notes:* *** p<0.01, ** p<0.05, * p<0.1

The regression results, however, are significant during the individual waves. During the March 02 to June 07, 2020, the adjusted $R^2$ increases to 6.1% and we observe a highly significant negative correlation (p-value <0.0005) of rural populations and the case-sentiment correlation. Therefore, as the variable "percent of the population living in rural areas" of a county increases, the correlation between COVID-19 cases and sentiment becomes more negative, which is expected. We see a similar correlation during the third wave from August 26 to December, 2020 albeit not quite as significant at a p-value of 0.005. Interestingly, during the second wave, the explanatory variable "percent of the population living in rural areas" remains highly significant, at a p-value <0.0005 but the signum of this correlation is reversed during this period. Instead of having a highly significant negative correlation, we observe a highly significant positive correlation during this period. Thus, during this time, rural areas are more likely to have a high tweet sentiment when having a higher number of COVID-19 cases which is not the expected behaviour. Looking at other population variables, we observe no significant link (at a level of significance of 1%) for any of the other population variables.

Towards understanding where we can observe a strong correlation between COVID-19 cases and tweet sentiment, this regression analysis shows a significant link to rural areas. But an unanswered question remains why the direction of the significant correlation changed in

rural (and, dually, non-rural) areas. If we could explain and predict this correlation for future pandemics, we could possibly use sentiment as an early-warning system for cases.

## Discussion and conclusions

Globally, across all counties of the United States and across the entire 16-months study period, we are not able to find a significant link between tweet sentiment and COVID-19 cases. Thus, we extended our analysis to spatial and temporal dimensions to investigate the spatial and temporal stationarity of such a trend and whether variations in the magnitude and direction of the correlation could be found at certain times or in certain locations. Our temporal analysis showed significant temporal patterns, including an interval of strong and significant negative correlation in the Summer of 2020. We also observe periods of strong and significant positive correlation (contrary to the expectation that high case numbers would align with lower sentiment). In both cases, the strong positive and negative correlations indicate that sentiment could be used as an indicator of COVID-19 cases.

In addition to temporal analysis, we also performed a spatial analysis, to understand whether different counties across the United States may also exhibit non-stationarity. We observed that areas in the Northeast of the U.S. exhibit a stronger negative correlation while East exhibits are stronger positive correlation. This result is also interesting, as it shows some regions of the U.S. may allow more accurate forecasting of infectious disease cases based on Twitter sentiment than others.

Finally, we performed a spatiotemporal analysis observing three distinct portions of our study period. Particularly the northeast United States oscillates between positive and negative correlation throughout the study period. These three results are quite interesting: They indicate that if the nature of the correlation—whether positive, negative, strong, weak, spatial, or temporal—can be understood during an infectious disease outbreak, tweet sentiment might be effectively used as a localized indicator.

For example, we find in the first wave of the COVID-19 pandemic that rural areas have a stronger negative correlation between COVID-19 cases and sentiment (as expected). Inversely, we might assume that urban areas will thus have a stronger positive correlation between COVID-19 cases and sentiment. This may mean that during the first wave of the COVID-19 pandemic, positive sentiment in urban areas and negative tweet sentiment in rural areas could be signals for higher COVID-19 cases. While we were able to get some insights into some underlying demographic factors that may explain the spatial patterns in our analysis, the models have a low R-squared, meaning that further investigation using other variables that could explain these patterns is needed. The observed spatial and temporal variability (and likely differences across spatial scales) may explain why a scientific consensus about the relationship between sentiment and COVID-19 disease prevalence has not been made.

We note that social media data from platform X is not without its limitations. One issue is the inherent bias and lack of representativeness in social media data like platform X. While this bias has not been quantified in the TBCOV dataset, other researchers have attempted to understand such bias in platform X data as a whole. For example, Mislove et al. [39] find that platform X users with geolocated Tweets represent a small fraction of the population (1.15%) and that populous counties are overrepresented. They also find a bias towards male users and that over/undersampling of different races of users is a function of location. This could affect our interpretation of the results, which may not reflect the relationship between sentiment and COVID-19 cases in a way that is representative of the broader population. In addition to the bias in the user base, high-profile events throughout the COVID-19 pandemic (e.g. major changes in policy, new public health interventions) as well as media-driven narratives

may disproportionately shape sentiment in the TBCOV dataset, potentially diverging from broader public sentiment and contributing to the observed spatial and temporal variation. Finally, misinformation, fake news, and partisanship on platform X may also explain why social media, specifically sentiment, may fall short as a consistent indicator for COVID-19 outbreaks [40,41]. For example, platform X sentiment has been found to correlate with public opinion about how the pandemic is being managed [42,43] and may not fully reflect case numbers or disease severity. The two are difficult to disentangle. Given the limitations of both case data from official sources and social media data, a more comprehensive disease detection surveillance system would leverage both.

Factors such as regional cultural norms, socio-political climates, and local pandemic responses could further shape the variability in the correlation between sentiment and COVID-19 cases. Further research is needed to understand both temporal and spatial processes that cause the correlations between Tweet sentiment and infectious diseases to shift across space and time. If we could understand these processes, we might be able to better identify when and where Tweet sentiment could be used as a signal for early disease outbreak warning.

## Author contributions

**Conceptualization:** James Stassinos, Taylor Anderson, Andreas Züfle.

**Data curation:** James Stassinos.

**Formal analysis:** James Stassinos.

**Funding acquisition:** Taylor Anderson, Andreas Züfle.

**Investigation:** James Stassinos, Andreas Züfle.

**Methodology:** James Stassinos, Taylor Anderson, Andreas Züfle.

**Project administration:** Taylor Anderson, Andreas Züfle.

**Supervision:** Taylor Anderson, Andreas Züfle.

**Validation:** James Stassinos.

**Visualization:** James Stassinos.

**Writing – original draft:** James Stassinos, Taylor Anderson, Andreas Züfle.

**Writing – review & editing:** James Stassinos, Taylor Anderson, Andreas Züfle.

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
