## [Decision Letter · Decision Letter 0]

4 Dec 2024

PONE-D-24-46561Towards using Tweet Sentiment for Infectious Disease DetectionPLOS ONE

Dear Dr. Anderson,

Thank you for submitting your manuscript to PLOS ONE. After careful consideration, we feel that it has merit but does not fully meet PLOS ONE’s publication criteria as it currently stands. Therefore, we invite you to submit a revised version of the manuscript that addresses the points raised during the review process.

We look forward to receiving your revised manuscript.

Kind regards,

Li-Pang Chen

Academic Editor

PLOS ONE

Journal Requirements:

4. Please update your submission to use the PLOS LaTeX template. The template and more information on our requirements for LaTeX submissions can be found at http://journals.plos.org/plosone/s/latex.

“This study was funded by the National Science Foundation (NSF) Awards #2109647, #2302968, and #2302970.”

“This study was funded by the National Science Foundation (NSF) Awards #2109647, #2302968, and #2302970.

Additional Editor Comments (if provided):

The manuscript has been reviewed by two referees. One of both provided a positive feedback while the other did not. Provided that your work is interesting, I decide to stand by the positive position and give the authors an opportunity to revise the manuscript. Hence, please carefully address all comments raised by two referees.

Reviewers' comments:

Reviewer's Responses to Questions

**Comments to the Author**

1. Is the manuscript technically sound, and do the data support the conclusions?

Reviewer #1: Partly

Reviewer #2: Yes

2. Has the statistical analysis been performed appropriately and rigorously? 

Reviewer #1: No

Reviewer #2: Yes

3. Have the authors made all data underlying the findings in their manuscript fully available?

Reviewer #1: Yes

Reviewer #2: Yes

4. Is the manuscript presented in an intelligible fashion and written in standard English?

Reviewer #1: Yes

Reviewer #2: Yes

5. Review Comments to the Author

Reviewer #1: # Overview

The paper aims to build on existing studies that find weak or mixed correlation between COVID incidence and Twitter sentiment in the United States by investigating the spatial and temporal variation in this correlation. To this end, the authors combine data from the TBCOV Tweet dataset with data on COVID prevalence from the CDC to generate average sentiment and COVID prevalence at the county–week level. They then calculate Pearson's correlation statistic on different subsets of this data.

Below, I give a high-level assessment of (a) the framing and literature review and (b) the methodology used. I then list some smaller issues that I encountered in the manuscript.

# Framing and literature

An initial issue with the manuscript is a mismatch between the framing and the analysis. The authors situate the importance of their analysis in terms of contributing to the rapid detection (or even prevention) of epidemic outbreaks. However the analysis itself does not provide any treatment of the the predictive value of sentiment for disease outbreaks. One of the main features of the relationship between COVID cases and Tweet sentiment illustrated in the paper is that there is a large amount of unexplained variability in this relationship across space and time. Because of this, the utility of the findings is overstated.

The literature that the authors cite at the beginning of the article does a good job motivating the potential for social media in epidemiological studies in general. However the paper overstates the novelty of the spatial analysis ti presents. Notably, the original paper accompanying the TBCOV dataset the paper relies on includes a similar county-level analysis.

I was also surprised that the authors did not reference the considerable body of research demonstrating systematic geographic variation in social media sentiment in the United States (e.g. Mitchell et al 2013). Given the emphasis on geographic variation in the analysis, such work seems particularly relevant.

# Methodology

Overall, the methodology used in the paper is well described, but it is overly simplistic for an investigation of the relationship between social media sentiment and disease prevalence. Among the main findings of the paper is that correlations between sentiment and COVID cases are often stronger when they are calculated on smaller geographically and temporally restricted slices of the data. This kind of finding should not be surprising, though: stronger correlations are exactly what one would expect if the data has any sort of underlying smoothness (i.e. temporal and spatial autocorrelation).

That said, the authors note some intriguing patterns such as the strong negative correlation from July to September of 2020 or the East–West differences in direction and clustering of correlation. Unfortunately, the authors do little to try to explain these patterns. Are the temporal (or geographic) patterns associated with national (or local) policy? Do the spatial patterns correspond with population density, socioeconomic differences, or medical resources? If the stated goal is to contribute to early warnings for outbreaks, explaining the differences in magnitude and direction of correlation is vital. In my view, a regression analysis that takes into account spatial and geographic features would be a much more appropriate method to use.

# Smaller issues

- In Figure 1(b), the rolling 7-day average of COVID cases per 100K in Washington falls below zero at the beginning of 2021. This warrants explanation (a quick view of the cited COVID data source doesn't show any negative values)

- On page 4, the authors state "Deriving the sentiment of each such set provides us with a function S(w, r) that returns the sentiment of all tweets observed in region r during week w." I couldn't find a description of how sentiment was aggregated across sets of tweets.

- In figure 2(b) and in the text on pages 6 and 7 the authors use p-values in a loose way to assess significance. A statistic like a Z-score would be more appropriate.

- On page 6, the authors state that "we observe that our hypothesis of temporal heterogeneity is supported," but relevant the hypotheses put forward on page 4 was that the correlation would be "stable over time."

- I do not believe the authors ever justified their choices of dates separating the three 'waves' used in the final spatiotemporal analysis.

References

Mitchell L, Frank MR, Harris KD, Dodds PS, Danforth CM (2013) The Geography of Happiness: Connecting Twitter Sentiment and Expression, Demographics, and Objective Characteristics of Place. PLoS ONE 8(5): e64417. https://doi.org/10.1371/journal.pone.0064417

Reviewer #2: Towards Using Tweet Sentiment for Infectious Disease Detection

The authors explore whether social media sentiment, specifically derived from tweets, can function as an early warning system for infectious disease outbreaks, focusing on COVID-19. Utilizing a dataset of 56,755,894 tweets from the platform X, they analyze spatial, temporal, and spatiotemporal correlations between tweet sentiment and COVID-19 case counts, aggregating data by county and week. Their temporal analysis identifies a significant negative correlation between sentiment and cases from June to September 2020, suggesting that declining sentiment could predict outbreaks. Spatially, they find contrasting trends: a negative correlation in the Eastern U.S. and a positive correlation in the Western U.S., highlighting regional variability. Spatiotemporal analysis uncovers even stronger correlations during specific periods and regions, emphasizing the potential of social media as a complementary tool to traditional public health surveillance systems if contextual dynamics are better understood.

The paper is well-structured and employs a consistent methodology. However, there are several areas where the authors could improve their work before publication:

1. Literature on Complex Systems and Social Media Communication

The manuscript would benefit from incorporating insights from complex systems research that explore the relationship between social media communication patterns and disaster prediction, prevention or even management after. For instance, studies like “Rapid Assessment of Disaster Damage Using Social Media Activity” could be relevant.

2. Critique of Traditional Data Collection Methods

While the authors critique testing bias and the lack of standardized reporting metrics in traditional methods, additional points could strengthen this discussion. For example, official census data often relies on diverse methodologies and is influenced by budgetary constraints, while surveys and questionnaires can introduce subjectivity in evaluation criteria. In the specific context of COVID-19 mapping data, studies such as “Geospatial Analysis and Mapping Strategies for Fine-Grained and Detailed COVID-19 Data with GIS” highlight these issues and could provide a solid foundation to justify the approach taken in this study.

3. Reliability of Social Media Data During Pandemics

The authors argue that microblogs like tweets are effective in detecting disease prevalence (e.g., influenza, dengue). However, why did such globally accessible and rapid communication channels fall short during the COVID-19 pandemic? This question warrants exploration to address the limitations of social media in large-scale health crises.

4. Contradictory Results in the Literature

The manuscript references findings like those by “Valdez et al were surprised to observe a positive correlation between US wide COVID-19 related Tweet sentiment and cases and deaths, meaning that as cases and deaths increase in the US, sentiment towards COVID-19 trends positive. This contradicts what would intuitively be expected. In another example, Feng and Kirkley find a weak negative or absent correlation between state-level Tweet sentiment and COVID-19 and cases and deaths”. How can these contradictory and counterintuitive results across studies be interpreted? A discussion of these inconsistencies in the literature would enhance the paper’s depth.

5. Complexity of Spatial and Temporal Contexts

The authors should consider the temporal and spatial heterogeneity of the pandemic’s impact. Policies and interventions varied significantly across counties and states, influencing social behavior and pandemic outcomes. For instance, Maleki (2022) examines social factors influencing compliance with COVID-19 interventions in the U.S., offering valuable insights into behavioral changes during the pandemic.

6. Matching Real-World and Social Media Populations

A key limitation of using Twitter data is the demographic bias of its user base, which tends to be younger, more educated, and urban. How well does this population represent broader societal trends? While some areas may align more closely with Twitter sentiment, others may diverge significantly. The authors should consider a twofold discussion:

(a) The replicability of real-world events in digital spaces, and

(b) The complexity of social space, where factors like heterogeneity, scale, and context introduce variability in data interpretation (e.g., Morales et al., 2022).

7. Expanding Discussion and Results

The discussion and results sections are currently underdeveloped. While the authors present key findings, they provide limited analysis or interpretation. Expanding these sections with a deeper examination of results, including comparisons with previous studies and a critical evaluation of implications, would significantly enhance the paper’s impact.

6. PLOS authors have the option to publish the peer review history of their article (what does this mean?). If published, this will include your full peer review and any attached files.

Reviewer #1: No

Reviewer #2: No

---

## [Author Response · Author response to Decision Letter 1]

10 Feb 2025

We have uploaded a Response to Reviewers as a pdf in the file attachments.

---

## [Decision Letter · Decision Letter 1]

8 May 2025

Towards using Tweet Sentiment for Infectious Disease Detection

PONE-D-24-46561R1

Dear Dr. Anderson,

We’re pleased to inform you that your manuscript has been judged scientifically suitable for publication and will be formally accepted for publication once it meets all outstanding technical requirements.

Kind regards,

Li-Pang Chen

Academic Editor

PLOS ONE

Additional Editor Comments (optional):

Reviewers' comments:

Reviewer's Responses to Questions

**Comments to the Author**

1. If the authors have adequately addressed your comments raised in a previous round of review and you feel that this manuscript is now acceptable for publication, you may indicate that here to bypass the “Comments to the Author” section, enter your conflict of interest statement in the “Confidential to Editor” section, and submit your "Accept" recommendation.

Reviewer #2: All comments have been addressed

2. Is the manuscript technically sound, and do the data support the conclusions?

Reviewer #2: Yes

3. Has the statistical analysis been performed appropriately and rigorously? 

Reviewer #2: Yes

4. Have the authors made all data underlying the findings in their manuscript fully available?

Reviewer #2: Yes

5. Is the manuscript presented in an intelligible fashion and written in standard English?

Reviewer #2: Yes

6. Review Comments to the Author

Reviewer #2: Dear Authors,

After the review conducted by the authors in this review round, the modifications have been satisfactorily addressed. For that, I can suggest that this manuscript can be accepted for publication.

Congrats on your achievement,

Best, the reviewer

7. PLOS authors have the option to publish the peer review history of their article (what does this mean?). If published, this will include your full peer review and any attached files.

Reviewer #2: No

---

## [Editor Report · Acceptance letter]

PONE-D-24-46561R1

PLOS ONE

Dear Dr. Anderson,

I'm pleased to inform you that your manuscript has been deemed suitable for publication in PLOS ONE. Congratulations! Your manuscript is now being handed over to our production team.

Kind regards,

on behalf of

Dr. Li-Pang Chen

Academic Editor

PLOS ONE